# Microbiome of Invasive Tick Species *Haemaphysalis longicornis* in North Carolina, USA

**DOI:** 10.3390/insects15030153

**Published:** 2024-02-24

**Authors:** Loganathan Ponnusamy, Nicholas V. Travanty, D. Wes Watson, Steven W. Seagle, Ross M. Boyce, Michael H. Reiskind

**Affiliations:** 1Department of Entomology and Plath Pathology, North Carolina State University, Raleigh, NC 27695, USA; travanty@gmail.com (N.V.T.); wwatson@ncsu.edu (D.W.W.); mhreiski@ncsu.edu (M.H.R.); 2Department of Biology and Southern Appalachian Environmental Research and Education Center, Appalachian State University, Boone, NC 28608, USA; seaglesw@appstate.edu; 3111 Mason Farm Road, MBRB 2336, Chapel Hill, NC 27599, USA; ross_boyce@med.unc.edu

**Keywords:** *Haemaphysalis longicornis*, developmental stage, microbiome, 16S rRNA, *Coxiella* spp.

## Abstract

**Simple Summary:**

The Asian longhorned tick (ALHT), *Haemaphysalis longicornis*, is an invasive pest that threatens domestic livestock. Normally found in Asia and the Pacific islands, where it is a vector of human disease, this tick was reported for the first time in the United States in 2017. In this study, we collected *H. longicornis* ticks of different developmental stages and used bacterial 16S rDNA amplicon sequencing to examine their microbiome. We identified numerous bacterial taxa, with *Coxiella*, *Sphingomonas*, *Staphylococcus*, *Acinetobacter*, *Pseudomonas*, *Sphingomonadaceae*, *Actinomycetales,* and *Sphingobium* as the most prevalent in the bacterial community. We documented a remarkable turnover in bacterial assemblage between life stages. These findings reveal important associations between life stages and their bacterial community and provide important insights to guide future research.

**Abstract:**

Ticks are one of the most important vectors of human and animal disease worldwide. In addition to pathogens, ticks carry a diverse microbiota of symbiotic and commensal microorganisms. In this study, we used next-generation sequencing (NGS) to survey the microbiomes of *Haemaphysalis longicornis* (Acari: Ixodidae) at different life stages collected from field populations in North Carolina (NC), USA. Sequence analyses were performed using QIIME2 with the DADA2 plugin and taxonomic assignments using the Greengenes database. Following quality filtering and rarefaction, the bacterial DNA sequences were assigned to 4795 amplicon sequence variants (ASVs) in 105 ticks. A core microbiome of *H. longicornis* was conserved across all ticks analyzed, and included bacterial taxa: *Coxiella*, *Sphingomonas*, *Staphylococcus*, *Acinetobacter*, *Pseudomonas*, Sphingomonadaceae, Actinomycetales, and *Sphingobium*. Less abundant bacterial taxa, including *Rickettsia* and *Aeromonas,* were also identified in some ticks. We discovered some ASVs that are associated with human and animal infections among the identified bacteria. Alpha diversity metrics revealed significant differences in bacterial diversity between life stages. Beta diversity metrics also revealed that bacterial communities across the three life stages were significantly different, suggesting dramatic changes in the microbiome as ticks mature. Based on these results, additional investigation is necessary to determine the significance of the *Haemaphysalis longicornis* microbiome for animal and human health.

## 1. Introduction

Worldwide, arthropod vectors play a vital role in the transmission of numerous infectious diseases, causing significant morbidity and mortality [1], with ticks, assassin bugs, fleas, blackflies, and mosquitoes as the most important groups [2]. Ticks are second only to mosquitoes in the disease burden they cause by transmitting infectious microbial agents, including causative agents of Lyme disease, spotted fever group (SFG) rickettsiosis, Babesiosis, human granulocytic anaplasmosis, and human monocytic ehrlichiosis, as well as several arboviruses [3,4].

Among ticks, the Asian longhorned tick, *Haemaphysalis longicornis* Neumann, 1901 (Ixodida: Ixodidae), transmits many human and animal pathogens. Historically endemic to eastern Asia, Australia, and New Zealand, this tick poses a major threat to the cattle industry and heavy infestations can lead to significant economic losses [5]. In Asia, *H. longicornis* is associated with the transmission of several pathogenic bacteria, protozoa, and viruses, namely *Anaplasma* spp., *Babesia* spp., *Borrelia* spp., *Rickettsia* spp., *Theileria* spp., and severe fever with thrombocytopenia syndrome virus [6,7,8,9]. In its invasive range in North America, *H. longicornis* tick is considered a potential vector of Powassan virus, which infects humans and animals (snowshoe hares and woodchucks) in the U.S. and Canada [10,11]. More recently, *Anaplasma phagocytophilum*, *Borrelia burgdorferi*, *Theileria orientalis*, and Bourbon virus have been detected in *H. longicornis* [12,13,14,15]. It is noted that the presence of pathogens may not translate into transmission in all cases [16,17], but a recent study demonstrated that laboratory-raised *H. longicornis* can acquire and transmit *R. rickettsii*, the causative agent of RMSF [18].

In North America, *H. longicornis*, was identified for the first time on a sheep in New Jersey, USA, in 2017 [7]. As of 13 October 2023, *H. longicornis* was reported in 19 states, and its range appears to be growing in more states in the USA [19,20,21]. In the USA, *Haemaphysalis longicornis* has been reported on humans, domestic animals (cattle, cats, dogs, goats, horses, sheep), and various wildlife (coyotes, gray foxes, groundhogs, opossums, raccoons, white-tailed deer) [19,22,23]. While evidence proving that this tick has the ability to feed on a broad range of host animals in the USA is increasing, its potential to act as a vector for human pathogens in the USA is still poorly understood.

Over the past decade, the employment of next-generation sequencing (NGS) has facilitated the discovery of novel tick-borne pathogens [24,25,26]. Recent research has demonstrated that the resident bacterial community of ixodid ticks can influence the reproductive fitness, physiological processes, and the acquisition, establishment, and transmission of certain tick-borne pathogens [27,28]. Further, a study of the microbiome of field-collected *H. longicornis* found several potential pathogens, including *Anaplasma bovis*, *Coxiella burnetii*, and *R. rickettsii* [29]. These findings demonstrate the need to investigate the bacterial community in the invasive *H. longicornis* ticks present in the USA to aid in identifying potential tick-borne pathogens.

The goal of this study is to investigate the field-collected bacteria community in *H. longicornis* at all three life stages in NC using NGS methods, exploring the impact of life stage on the microbiome, and provide a foundation for future studies illustrating the biological significance of bacteria in this tick species.

## 2. Materials and Methods

### 2.1. Tick Collection and Identification

Ticks were collected from five different counties in the Mountain and Piedmont regions of NC (Appendix A) by dragging for one hour or for a minimum of 600 m^2^ within each area using a 1 m^2^ cotton-flannel cloth. In addition, ticks were also collected directly from cattle. Ticks collected from October 2019 through September 2021 were stored in 95% ethanol. All ticks were identified by sex, life stage, and species using taxonomic keys [30]. Tick identifications were confirmed by the National Veterinary Services Laboratory, Ames, Iowa, and reported in the Asian longhorned tick situation report (USDA, Washington, DC, USA).

### 2.2. DNA Extraction and Library Construction

One hundred and eleven tick samples, including 38 adults (all female, individually), 48 nymphs (individually), and 25 larvae pools (five/pool) were prepared and DNA extracted. To remove surface contaminants (including microbes on the surface of a tick) prior to extraction, each individual tick or pool was washed using previously described methods [31]. After surface sterilization, the ticks were then transferred individually (females and nymphs) or pooled (larvae) into sterile 2 mL screw-capped microcentrifuge tubes containing 10 to 12 sterilized 3 mm glass beads (Cat. 11-312A, Fisher Scientific, Hampton, NH); female (referred to hereafter as an adult) and nymphal ticks were then cut into sections with a sterile scalpel into QIAGEN ALT buffer. Next, samples were homogenized in a FastPrep FP120 cell homogenizer (Thermo Electron Corporation, Waltham, MA, USA). From each sample, DNA isolation was performed using the QIAGEN DNeasy Blood and Tissue Kit (QIAGEN, Valencia, CA, USA) based on the manufacturers’ directions. The extracted DNA concentration were determined using NanoDrop 1000 (Thermo Scientific, Wilmington, DE, USA). The extracted total DNA was stored at −40 °C until further processing.

### 2.3. Bacterial 16S rRNA Gene Amplification and Library Preparation

Libraries of 16S rRNA amplicons were created from the 111 tick samples. PCR amplification and library preparation followed the Illumina 16S metagenomic sequencing protocol. Briefly, in each PCR, purified DNA from samples was used as template DNA, and using 341F/806R universal primers, the initial PCR amplified the hypervariable region (V3-V4, approximately 464 bp) of the bacterial 16S rRNA genes [32]. A negative control reaction was also performed using all reagents with DNA-free water serving as the template. Amplicons were purified using AmPure XP beads (Agencourt Bioscience Corporation, Beverly, MA, USA), followed by a second (index) PCR with dual indexes (i5 and i7) from the Illumina Nextera index kit V2 (Illumina, San Diego, CA, USA). PCR products (amplicons) were purified using AmPure XP beads, and DNA concentration was determined with Quant-iT PicoGreen (Molecular Probes, Inc. Eugene, OR, USA). Final libraries were pooled in equimolar amounts. Paired-end (PE) sequencing of the library was performed on Illumina Mi-Seq PE300 at the UNC Core Microbiome Facility at Chapel Hill, NC, USA.

### 2.4. Bioinformatics Data Processing and Statistical Analyses

All analyses were completed using the Quantitative Insights into Microbial Ecology (QIIME2) pipeline [33]. Illumina FASTQ files were demultiplexed and quality-filtered, and reads were then denoised, paired-end reads merged, and chimeric sequences were deleted using the DADA2 plugin [34]. Amplicon sequence variants (ASVs) were aligned with the Align-to-tree-mafft-fasttree pipeline [35], and a phylogeny of the ASVs was constructed with q2-phylogeny [36]. Taxonomic classification of ASVs was performed on representative unique sequences that were generated from DADA2 using a trained Bayes classifier from Greengenes 13_8 with 99% sequence similarity of the OTU data set [37].

To ensure an even sampling depth, all the samples were rarefied to 4800 sequences. A consensus-rarefied ASV table was used to determine alpha diversity using the two different indices: (1) number of observed features (OFs) [38], which is a measure of species richness; (2) and the Shannon diversity index [39], which is an evenness metric based on the abundance of the observed taxa. Non-parametric Kruskal–Wallis tests (α = 0.05) were performed to determine the effects of stage of development (adults, nymphs, and larvae) on each alpha diversity metric; the effects of location collection (by county) on alpha diversity was also determined separately for each stage of development. Weighted UniFrac metrics [40] were used to measure beta diversity; the permutational multivariate analysis of variance (PERMANOVA) [41,42] tested plot distances’ output data from the beta-group-significance visualizer in plugin EMPeror [43] to determine group differences. Bacterial taxa having differential abundances dependent on the developmental stage of *H. longicornis* were identified using ANCOM [44,45] as part of the QIIME2 pipeline.

### 2.5. Phylogenetic Analyses of Aeromonas, Coxiella, Rickettsia, and Staphylococcus Sequences

Phylogenetic trees were created to confirm and/or improve the phylogenetic relationships for *Aeromonas*, *Coxiella*, *Rickettsia,* and *Staphylococcus* sequence variants from this study with other closely related sequences obtained from the NCBI database by BLASTn analysis (accessed on 1 December 2023). Multiple alignments were performed with ClustalW program [46]. For each genus, separate phylogenetic trees were constructed using the neighbor-joining (NJ) analyses with the Kimura two-parameter model in MEGA 11 software [47]. Bootstrapping percentage (1000 re-sampling iterations) were calculated for NJ trees.

## 3. Results

### 3.1. 16S rRNA V3-V4 Region Sequencing Results

After clustering, the removal of chimeras, and filtering, 1,686,038 sequences were retained for analyses. The median sequencing output was 16,100 reads. Based on the saturation plateau of the rarefaction curve (Appendix A), a sampling depth of 4800 was selected to allow for the retention of the greatest number of samples while capturing an adequate bacterial diversity for comparison between samples. Six samples with lower sequencing depth (less than 4800 sequencing reads) were removed from the diversity analyses. Following rarefaction, 4795 ASVs in 105 samples were assigned taxonomic identifications, and sequences identified as mitochondria or chloroplasts were subtracted from the analysis. Five hundred and two genera (level-6 classification) and 652 species (level-7 classification) were identified from the 105 ALT samples that were included in diversity analyses.

### 3.2. Alpha Diversity

The *Haemaphysalis longicornis* bacterial community richness and evenness was assessed on observed features (OFs). Kruskal–Wallis tests show that the number of OFs (ASVs) (*H* = 16.265, *p* < 0.001; Figure 1) and Shannon’s index (*H* = 36.4234, *p* < 0.0001; Figure 2) significantly differed across the life stages (adults, nymphs, and larvae); results are summarized in Table 1. Pairwise comparisons showed that nymphs have significantly lower observed feature richness compared to adults and larvae. All three life stages had significantly different Shannon’s diversity; larvae had the highest Shannon’s diversity, and nymphs had the lowest (Table 1). Testing for effects of collection location within stages showed that Shannon’s diversity in adults (*H* = 27.6429, *p* < 0.0001) and nymphs (*H* = 16.8412, *p* = 0.0007) were dependent on the county in which they were collected; Shannon’s diversity in larvae (*H* = 0.0277, *p* = 0.8678) was not dependent on area of collection. However, when looking at OF diversity, there was significant differences by location for larvae (*H* = 15.7300, *p* < 0.0001), nymphs (*H* = 22.5764, *p* < 0.0001), and adults (*H* = 25.9940, *p* < 0.0001). The number of collection locations within each stages of development and their respective OF and Shannon index values are summarized in Table 2.

### 3.3. Beta Diversity

Principal coordinate analysis of weighted UniFrac distances explained by each principal coordinate axis is represented in the corresponding axis label; axis 1 explains 48.92% of the variability, axis 2 explains 21.05% of the variability, and axis 3 explains 7.128% of the variability; together, the axes explain 77.09% of the variability (Figure 3). Bacterial communities across the three life stages were significantly different according to a PERMANOVA test (weighted *F* = 14.963, *p* < 0.001). Pairwise PERMANOVA comparisons revealed significant differences between adults and larvae (*F* = 7.210, *p* < 0.001), adults and nymphs (*F* = 9.144, *p* < 0.001), and larvae and nymphs (*F* = 32.820, *p* < 0.001), which indicated that the bacterial composition was significantly different between life stages (Appendix A). For each life stage, the county of capture had significant effects on bacterial community composition; larvae communities differed between Surry and Catawba counties (*F* = 20.453, *p* < 0.001); nymph communities were dependent on county (Surry, Catawba, Buncombe, and Madison) (*F* = 5.497, *p* < 0.001), as well as adult communities (Ashe, Surry, and Catawba) (*F* = 15.698, *p* < 0.001).

### 3.4. Bacterial Community Composition

By 16S rRNA gene sequencing, the most abundant genera across all ticks analyzed (>2% of the total count) were *Coxiella* (42.87%), *Sphingomonas* (6.64%), *Staphylococcus* (5.02%), *Acinetobacter* (4.43), *Pseudomonas* (3.02%), Sphingomonadaceae (2.74%), Actinomycetales (2.50%), and *Sphingobium* (2.53%) (Figure 4). The genus *Coxiella* was the most prevalent bacteria found in adults (47.44%) and nymphs (72.26%) of *H. longicornis*, whereas *Sphingomonas* (12.45%) and *Acinetobacter* (11.43%) were abundant in larvae of *H. longicornis* (Figure 4). Low abundance of the pathogenic genus *Aeromonas* was found in larvae (*n* = 10) from Catawba, a nymph (*n =* 1) from Madison, and adults (*n =* 6) in Ashe counties. *Rickettsia* was also detected in a larva (*n =* 1) in Catawba County and nymphs (*n =* 10) in Surry County. The analysis of composition of microbiome (ANCOM) was performed to identify the taxa that were differentially abundant across life stages of *H. longicornis*. With the use of ANCOM, we found nine genus-level bacterial taxa with differential abundances between *H. longicornis* larvae, nymphs, and adults (Appendix A).

### 3.5. Phylogenetic Analyses

To further identify taxa of interest (including *Coxiella* spp., *Staphylococcus* spp., *Rickettsia* spp., and *Aeromonas* spp.), phylogenetic trees (via neighbor-joining) were created from ASVs classified as these genera and sequences recovered from GenBank with the highest similarities.

Because *Coxiella* spp. include known pathogens, we compared our 31 ASVs that were identified in the genus *Coxiella*, with the five closest related species from BLASTn searches to create phylogenetic trees based on NJ (Appendix A). The phylogenetic analysis showed that 25 of the *Coxiella* ASVs clustered (99 to 100% similarity) with *Coxiella* isolate HT-131 (KT835656), an endosymbiont of *Haemaphysalis longicornis* from Korea. The phylogeny revealed two additional distinct clusters of four ASVs and two ASVs, which we considered as putative novel species.

We constructed a *Staphylococcus* phylogeny (data not shown), including 58 ASVs classified as *Staphylococcus* spp. and the 12 sequences from GenBank with the highest similarity. We found only 66% (36/58) of ASVs that met the genus threshold of 94.5% for the 16S rRNA gene nucleotide identity. These ASVs were matched (98.5 to 100% similarity) to different species in *Staphylococcus*, including known pathogens, e.g., *Staphylococcus aureus* and *Staphylococcus haemolyticus*, *S. saprophyticus*, and *S. xylosus*.

We built a phylogeny of *Rickettsia* composed of three ASVs classified as *Rickettsia*, and 16 sequences from GenBank having the highest similarities (Appendix A). One of the *Rickettsia* ASVs clustered to *Rickettsia amblyommatis* isolate An13 (CP015012) with a 100% similarity, while two ASVs clustered separately.

Three ASVs classified as *Aeromonas*, and nine sequences from GenBank with the highest similarities were used to construct phylogeny (Appendix A). One of the *Aeromonas* ASVs clustered to *Aeromonas molluscorum* and *Aeromonas encheleia* with a 100% similarity; another ASV clustered to *Aeromonas salmonicida* subsp *masoucida* with a 100% similarity, and another ASV formed an independent lineage, which we consider as a putative novel species.

## 4. Discussion

The purpose of this study was to identify the patterns of tick-associated microbial communities and their diversity in invasive *H. longicornis* collected in NC, the state with some of the highest reported rates of SFGR and ehrlichiosis in the USA [48,49,50]. We did not find evidence that *H. longicornis* in NC harbored known, endemic pathogens, although we did clearly demonstrate closely related taxa in these ticks.

We also found that ticks in NC had different microbial diversities by life stage, with nymphs having the lowest microbial diversity. In contrast, a previous study on *H. longicornis* ticks from Korea detected a higher microbial diversity for nymphs than adult ticks [29]. However, inconsistent results have also been reported regarding the dynamics of bacterial diversity across the *Ixodes* stage. For example, previous studies by Kwan et al. [51] and Swei and Kwan [52] showed decreasing bacterial diversity during tick development, while Zolnik et al. [53] documented increasing diversity. We also noted that collection site had an effect on diversity, supporting previous research showing that *Ixodes* tick microbiomes differ from place to place [54,55]. Many studies have reported different bacterial community structures in ticks by sample collection site, presence of host blood, and degree of engorgement [31,52,54,56,57]. Further, the source of blood meal and host identity were also found to significantly impact the microbial population within the body of arthropods, something we were not able to address in this study [58,59].

Overall, we found several bacterial genera with an abundance of more than 2.0% associated with the *H. longicornis* tick: *Coxiella*, *Sphingomonas*, *Staphylococcus*, *Acinetobacter*, *Pseudomonas*, *Sphingomonadaceae*, *Actinomycetales,* and *Sphingobium*. Of all genera detected in this study, *Coxiella* spp. was found to be over-represented in all larval, nymphal, and adult *H. longicornis* (present in 100% of the samples). Interestingly, our results indicated that nymphs (72.26%) tended to have high proportions of sequences of *Coxiella* symbionts more frequently than larval and adult ticks (8.90% of larvae and 47.45% of adults studied). These maternally inherited *Coxiella*-like endosymbionts, highly prevalent in soft and hard ticks, have been reported to be involved in mutualistic interactions with their arthropod hosts [60]. These bacteria are genetically similar to *C. burnetii*, which causes the zoonosis Q fever and are distributed worldwide [61]. Previous studies have shown that the agent is transmitted horizontally from ticks (e.g., *Haemaphysalis humerosa*, *H. bispinosa*, and *Rhipicephalus sanguineus*) to mammals [62]. *Coxiella* endosymbiont bacteria have been reported in several tick species, such as *H. lagrangei*, *H. hystricis H. obesa*, and *H. shimoga* [63]. In addition, *Amblyomma americanum*, *Rhipicephalus sanguineus*, *Ixodes uriae*, and the soft tick *Ornithodoros rostratus* [64,65,66] reported the presence of *Coxiella* endosymbiont. The highest prevalence was reported in the genus *Rhipicephalus* at 70% (53 out of 75), while it was lower in other genera, such as *Hyalomma* with 37% (26 out of 70). These endosymbionts are best known for their ability to transfer B vitamins to ticks along with cofactors and play an important role in tick growth, reproduction, and metabolism [28,66,67]. Furthermore, *Coxiella* endosymbiont might be responsible for human infections, causing symptoms such as neck lymphadenopathy syndrome and scalp eschar through tick bites [68]. Therefore, these *Coxiella* bacteria’s pathogenicity and zoonotic potential merit further investigation.

In addition, we found that the genus *Staphylococcus*, a potentially pathogenic genus that was the third most abundant taxa among all samples, had higher frequency in adults and larvae than nymphs. Ticks often carry *Staphylococcus* spp., which are generally found on the skin and mucous membranes of the ticks’ hosts [69,70,71]. Based on our phylogenetic tree, some of the ASVs are more closely related to *Staphylococcus aureus*, *S. haemolyticus*, *S. delphini*, *S. vitulinus*, *S. saprophyticus*, *S. xylosus*, *S. hominis*, and *S. croceilyticus*. Among these, some are known commensals and pathogens of mammals, for example, *S. aureus* is a common human commensal, but it can also cause invasive infections in many mammal species [72,73,74]. Some *Staphylococcus* species are pathogenic to tick vectors. For example, *Staphylococcus saprophyticus* and *S. xylosus*, which are found on bovine skin, can cause fully engorged adult ticks to lose the ability to oviposit and eventually lead to death [75].

We detected *Rickettsia* spp. in *H. longicornis* larvae (4.1% of tick larvae; 1/24) and nymphs (22%; 10/44). None were detected in adult ticks. Notably, the maternally inherited *Rickettsia* with lower prevalence in larvae than in nymphs could be due to the sensitivity of sequencing technology. A previous study by Travanty et al. [76] showed that extremely low-abundance bacteria like *Rickettsia* spp. are less sensitive in being detected with NGS than nested PCR that targets single-species detection. Phylogenetic analyses based on 16S rRNA gene sequences showed that the *Rickettsia* spp. detected in *H. longicornis* were *R. amblyommatis*. Across the world, 34 tick species in 17 countries have been identified as carrying *Rickettsia amblyommatis*, most of which are present in the USA. In NC, it was found that the most common exposure to *R. amblyommatis* in humans is following a bite from a lone star tick [77]. *Rickettsia amblyommatis* has consistently been found at high levels in *A. americanum,* a widely distributed human-biting tick in the USA [78,79,80]. For many years, there has been disagreement as to whether *R. amblyommatis* causes disease in mammals [81]. For instance, intravenously infected mice showed mild disease with clinical signs, including weight loss [82]. In a recent study, guinea pigs that were experimentally infected with this species showed no clinical signs, or even mild fever [83]. However, the guinea pigs also showed multiday scrotal edema and ear dermatitis. Therefore, this species may cause disease in some host(s) and needs to be closely monitored.

We detected the genus *Aeromonas* in 17.1% of tick samples. These bacteria are commonly found in numerous aqueous environments, including groundwater, drinking water, river water, irrigation water, seawater, and reclaimed wastewater [84,85,86]. Previous studies have found *Aeromonas* bacteria in ticks [87,88]. *Aeromonas* species have been determined to cause an extensive variety of diseases in humans and animals [89]. In recent years, *Aeromonas* species have been the focus of considerable research due to their potential to cause a range of maladies in individuals with compromised immune systems.

Among the remaining bacterial species, several are known to circulate in ticks, including *Sphingomonas*, *Acinetobacter*, *Pseudomonas*, Sphingomonadaceae, Actinomycetales, *Methylobacterium*, Enterobacteriaceae, *Mycobacterium*, *Macrococcus*, *Rhizobium*, Cedecea, *Streptococcus*, and *Brevundimonas* [90]. The diversity of these less observed taxa could correspond to variations in the environmental microbiota [91]. These genera could also be bacteria that ticks pick up from the external environment, such as when they take in water, or through their spiracles.

## 5. Conclusions

This study utilized a next-generation sequence analysis to conduct a comprehensive analysis of bacterial communities and to identify variations in bacterial communities present in the invasive *H. longicornis*. The results presented in this study provide a scientific foundation for reinforcing in-depth analysis of tick microbiome composition at the species level and for exploring the role of the identified microorganisms in ticks and control strategies for ticks and tick-related diseases in the United States.

## Figures and Tables

**Figure 1 insects-15-00153-f001:**
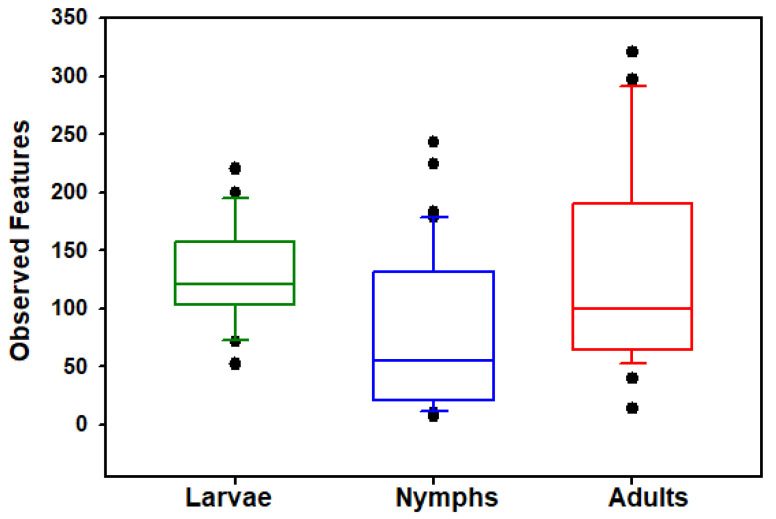
Comparisons of bacterial alpha diversity of larval, nymphal, and adult *Haemaphysalis longicornis*. Box plot data represent values from the specific OFs of larvae, nymphs, and adults Filled circles outside the horizontal line represent outliers.

**Figure 2 insects-15-00153-f002:**
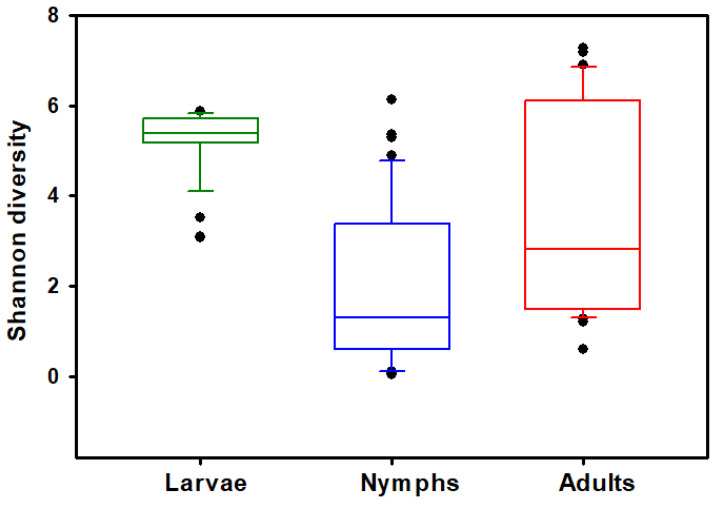
Comparisons of bacterial alpha diversity of larval, nymphal, and adult *Haemaphysalis longicornis*. Box plot data represent values from the specific Shannon’s entropy distributions of larvae, nymphs, and adults. Filled circles outside the horizontal line represent outliers.

**Figure 3 insects-15-00153-f003:**
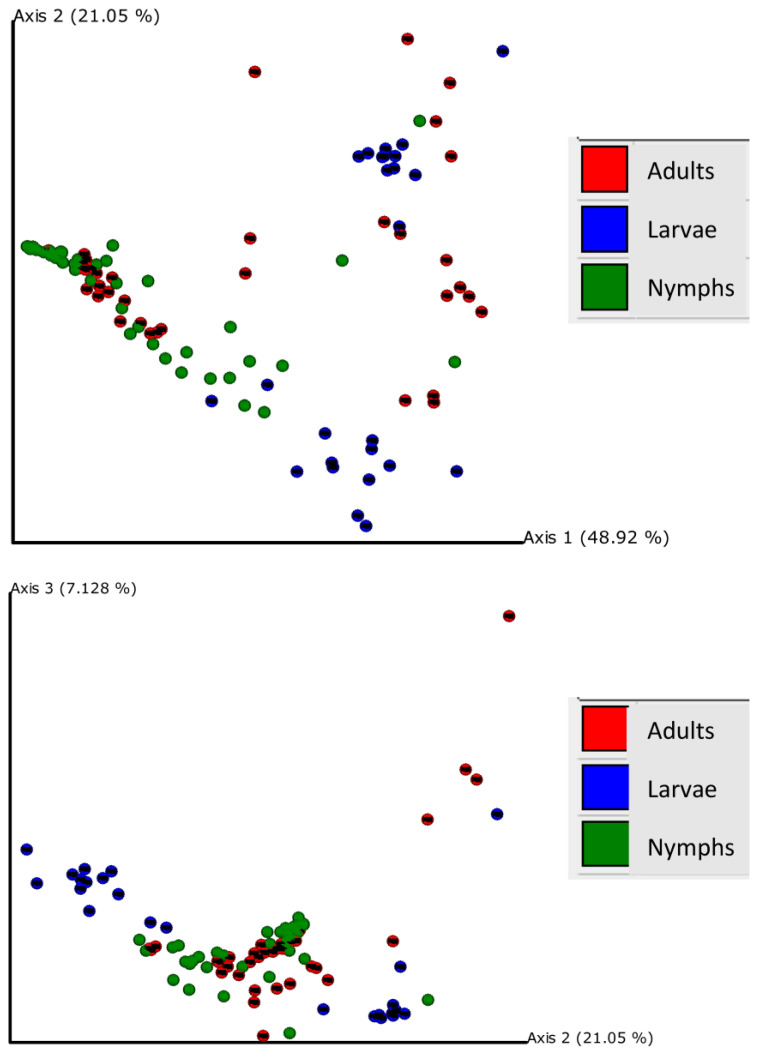
Principal coordinate analysis plot for larvae (blue), nymphs (green), and adults (red); distances across samples are calculated through weighted UniFrac distances.

**Figure 4 insects-15-00153-f004:**
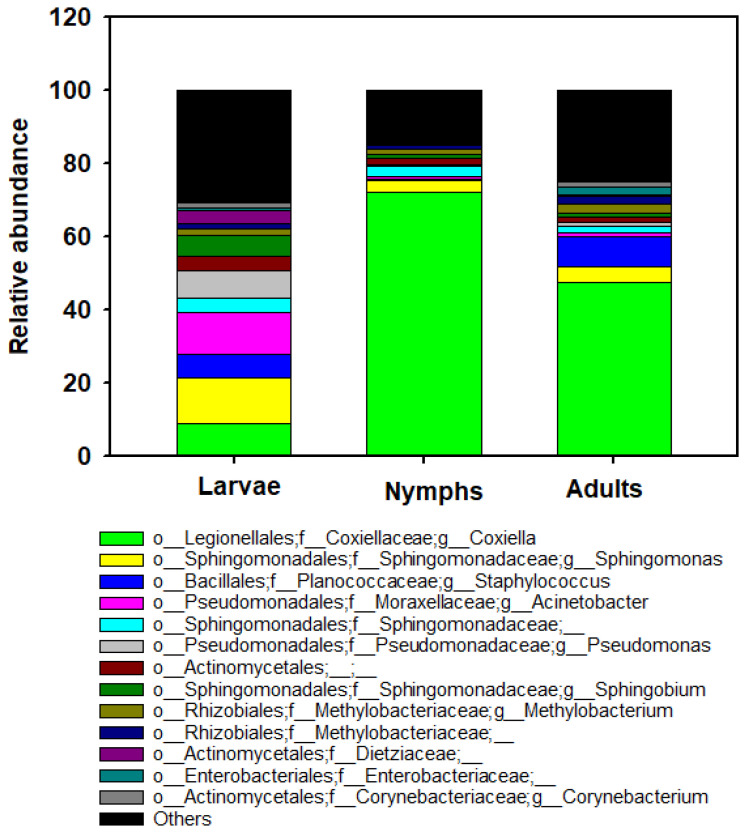
Bar graph depicting the bacterial amplicon sequence variants (ASVs) composition at the genus level for each stage. Other groups represent all taxa with relative abundance below 2%. Data represent average values from larval, nymphal, and adult ticks collected from NC.

**Table 1 insects-15-00153-t001:** Effects of the developmental stage of *Haemaphysalis longicornis* on microbiome alpha diversity.

Stage	*n*	Average (SE) Observed Features	Average (SE) Shannon’s Diversity
Adults	36	133.0 (14.2) a	3.608 (0.381) b
Nymphs	44	79.3 (9.9) b	2.037 (0.268) c
Larvae	25	129.6 (8.3) a	5.237 (0.137) a

a, b, c indicate pairwise groupings.

**Table 2 insects-15-00153-t002:** Effects of county on the microbiome alpha diversity of *Haemaphysalis longicornis*.

Stage	Counties	*n*	Average (SE) Observed Features	Average (SE) Shannon’s Diversity
Adults	Ashe	15	217.7 (16.0) a	6.113 (0.237) a
	Catawba	7	60.1 (4.4) b	1.339 (0.157) c
	Surry	14	78.6 (8.6) b	2.057 (0.184) b
Nymphs	Buncombe	12	52.4 (14.2) bc	1.242 (0.468) bc
	Catawba	10	163.6 (13.3) a	3.792 (0.270) a
	Madison	2	172.5 (4.5) ab	4.647 (0.709) ab
	Surry	20	44.0 (7.5) c	1.376 (0.317) c
Larvae	Catawba	10	168.8 (9.7) a	5.415 (0.076) a
	Surry	15	103.5 (6.0) b	5.118 (0.221) a

a, b, c indicate pairwise groupings within each stage of development.

## Data Availability

All raw sequence data generated during this study are available in the National Center for Biotechnology Information (NCBI) Sequence Read Archive (SRA) with the accession number under the BioProject SUB14135222. Data sequences are available at SRA data: PRJNA1061072.

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
