# Peer review of "Microbiome of Invasive Tick Species Haemaphysalis longicornis in North Carolina, USA"

_insects, 2024, doi:10.3390/insects15030153_

Round 1
Reviewer 1 Report
Comments and Suggestions for Authors
No edits. Well written paper that answered my questions as the text evolved. Thank you for the useful information.
Author Response
Comments: No edits. Well written paper that answered my questions as the text evolved. Thank you for the useful information.
Reply: Thank you for taking the time to review our Asian longhorned tick microbiome manuscript, and we sincerely appreciate the reviewer’s positive comments.
Reviewer 2 Report
Comments and Suggestions for Authors
Dear Authors, the proposal of the manuscript is very interesting, and with concise results. I made some minor corrections trought the text. Please pay attention to the comments highlighted in the PDF attached.

Comments on the Quality of English LanguageThat's fine. No major issues.
Author Response
Comments: Dear Authors, the proposal of the manuscript is very interesting, and with concise results. I made some minor corrections trought the text. Please pay attention to the comments highlighted in the PDF attached.
Reply: Thank you for the positive assessment of our paper. The comments listed in the PDF have been corrected in the revised manuscript, including the English language.
Comments: Line 48: Missing year of the authorship
Reply: Done
Comments: Lines 49-50: Suggestion: I believe that if this sentence were rewritten, it would be better to understand. The concept was well said, but sometimes it is truncated in the form of writing. Please pay attention through the text, this happen more times
Reply: This passage has been modified. The entire manuscript has been reviewed and edited to streamline the manuscript.
Comments: Line 52: change for Bacteria and Protozoa. In this ways, you are saying that all these microrganisms are pathogenic, and it is not true. There are some are endossymbionts.
Reply: As suggested, this change has been made.
Comments: Lines 63-65: In US? or in the original countries?
Reply: We agree; the suggested country details have been added.
Comments: Line 66: change 'for this tick'
Reply: Corrected.
Comments: Line 154: Please, correct the typo
Reply: This mistake has been corrected.
Comments: Line 166: Please, put the meaning.
Reply: Done.
Comments: Line 166: Please change in all the text for 'p' in lower case.
Reply: This has been corrected in the entire manuscript.
Reviewer 3 Report
Comments and Suggestions for Authors
Dear authors,
the manuscript entitled „ Microbiome of invasive tick species Haemaphysalis longicornis in North Carolina, USA” presents the result of a NGS study on the bacterial diversity within the named tick species.
The introduction gives the needed background information and highlighted the aims of the studies. Further, the used methods are properly described and the results are well presented and illustrated. The discussion categorizes the results and compares them with the relevant literature.
No further comments.
Author Response
Comments: the manuscript entitled „ Microbiome of invasive tick species Haemaphysalis longicornis in North Carolina, USA” presents the result of a NGS study on the bacterial diversity within the named tick species. The introduction gives the needed background information and highlighted the aims of the studies. Further, the used methods are properly described and the results are well presented and illustrated. The discussion categorizes the results and compares them with the relevant literature. No further comments.
Reply: We wish to thank the reviewer for favorable reviews.